# Lipid and Energy Metabolism of the Gut Microbiota Is Associated with the Response to Probiotic *Bifidobacterium breve* Strain for Anxiety and Depressive Symptoms in Schizophrenia

**DOI:** 10.3390/jpm11100987

**Published:** 2021-09-30

**Authors:** Ryodai Yamamura, Ryo Okubo, Noriko Katsumata, Toshitaka Odamaki, Naoki Hashimoto, Ichiro Kusumi, Jinzhong Xiao, Yutaka J. Matsuoka

**Affiliations:** 1Division of Biomedical Oncology, Institute for Genetic Medicine, Hokkaido University, Sapporo 060-0815, Japan; ryamamura@igm.hokudai.ac.jp; 2Department of Clinical Epidemiology, Translational Medical Center, National Center of Neurology and Psychiatry, Tokyo 187-8551, Japan; 3Next Generation Science Institute, Morinaga Milk Industry Co. Ltd., Zama 252-8583, Japan; n_katumt@morinagamilk.co.jp (N.K.); t-odamak@morinagamilk.co.jp (T.O.); j_xiao@morinagamilk.co.jp (J.X.); 4Department of Psychiatry, Hokkaido University Graduate School of Medicine, Sapporo 060-8638, Japan; hashinao@med.hokudai.ac.jp (N.H.); ikusumi@med.hokudai.ac.jp (I.K.); 5Division of Health Care Research, Center for Public Health Sciences, National Cancer Center Japan, Tokyo 104-0045, Japan; yumatsuo@ncc.go.jp

**Keywords:** gut microbiota, schizophrenia, depression, anxiety, probiotics, functional genes

## Abstract

A recent meta-analysis found that probiotics have moderate-to-large beneficial effects on depressive symptoms in patients with psychiatric disorders. However, it remains unclear how the baseline gut microbiota before probiotic administration influences the host’s response to probiotics. Therefore, we aimed to determine whether the predicted functional profile of the gut microbiota influences the effectiveness of probiotic treatment in patients with schizophrenia. A total of 29 patients with schizophrenia consumed *Bifidobacterium breve* A-1 (synonym *B. breve* MCC1274) for 4 weeks. We considered patients who showed a 25% or more reduction in the Hospital Anxiety and Depression Scale total score at 4 weeks from baseline to be “responders” and those who did not to be “non-responders”. We predicted the gut microbial functional genes based on 16S rRNA gene sequences and applied the linear discriminant analysis effect size method to determine the gut microbial functional genes most likely to explain the differences between responders and non-responders at baseline. The results showed that lipid and energy metabolism was elevated at baseline in responders (*n* = 12) compared to non-responders (*n* = 17). These findings highlight the importance of assessing the gut microbial functional genes at baseline before probiotic therapy initiation in patients with psychiatric disorders.

## 1. Introduction

The close relationship between the gut and the brain, termed the gut–brain axis, is supported by numerous basic and clinical studies showing that the gut microbiota influences the host’s mental state [1]. Probiotics, defined as “live microorganisms which when administered in adequate amounts confer a health benefit on the host”, have been attracting attention as a novel treatment for mental disorders. Probiotics such as *Bifidobacterium* and *Lactobacillus* were determined in a recent meta-analysis to have mild beneficial effects on depressive symptoms in patients with mental disorders [2]. In line with the results of this meta-analysis, we also reported the beneficial effects of *Bifidobacterium breve* A-1 on anxiety and depressive symptoms in patients with schizophrenia [3].

While probiotics are attracting attention, some researchers have focused on the influence of the gut microbiota on the host response to pharmacotherapy [4]. For example, the efficacy of immune checkpoint inhibitors for cancer depends on the patient’s gut microbiota [5]. Their anticancer effects are related to the relative abundance of *Bifidobacterium*, acting via augmented immune activity [6] and the amounts of metabolites produced by gut microbiota [7]. However, to our knowledge, it remains unclear how the baseline gut microbiota before probiotic administration influences the host’s response to probiotic therapy. In this context, using data from our previous interventional study [3], we sought to determine which predicted functional profiles of the gut microbiota at baseline are associated with improvement of anxiety and depressive symptoms. This functional gene profiling approach allowed us to clarify the function of the gut microbiota as a whole.

## 2. Materials and Methods

### 2.1. Study Design and Procedure

Our previous interventional study was conducted from November 2017 to May 2018 [3]. We recruited participants among consecutive outpatients with schizophrenia based on the following inclusion and exclusion criteria. The inclusion criteria were as follows: outpatients, aged 20 years or older, not hospitalized for at least 6 months since last discharge, and anxiety and depressive symptoms rated by doctors as ≥10 points on the Brief Psychiatric Rating Scale anxiety and depressive subscale (items 1, 2, 5 and 9).

The exclusion criteria were as follows: uncontrolled disease or untreatable malignancy; cognitive impairment or disorientation; severe suicidal ideation or symptoms requiring urgent treatment; desire to take medication for anxiety or depressive symptoms; antidepressant medication in the past month; daily consumption of foods or supplements containing Bifidobacterium; heavy alcohol consumption (>500 mL of beer/day); psychiatric disorders other than schizophrenia, mood disorders, or anxiety disorders; any other conditions deemed inappropriate by the physician in charge.

For the first 4 weeks, the participants consumed two 2-g sachets of freeze-dried *Bifidobacterium breve* A-1 (synonym *B. breve* MCC1274) per day, each containing 5.0 × 10^10^ colony-forming units. Fecal samples were collected from each patient prior to probiotic administration, and subjective anxiety and depressive symptoms were assessed using the self-administered Hospital Anxiety and Depression Scale (HADS) [8] every 4 weeks. Participants showing a 25% or more reduction in the HADS total score at 4 weeks from baseline were regarded as displaying a clinical response. Participants showing a clinical response were defined as “responders” and those not showing a response were defined as “non-responders”.

### 2.2. Bacterial DNA Extraction and Sequencing

Fecal bacterial DNA was extracted and purified as described previously [9]. We then amplified the V3–V4 region of bacterial 16S rRNA and sequenced it using the Illumina MiSeq platform (Illumina, San Diego, CA, USA) according to a previously described method [10].

### 2.3. Bioinformatics and Statistical Analysis

From trimming of the paired-end read FASTQ files obtained by 16S rRNA amplicon sequencing to analysis of gut microbiota diversity, all steps were carried out using QIIME 2. First, we demultiplexed the raw sequence results and used the Deblur algorithm to identify microbial operational taxonomic units (OTUs). The output feature table was diluted to 9000 sequences per sample. We then taxonomically classified the OTUs into 5 taxonomic rank categories—phylum, order, class, family, and genus—by using the SILVA 132 reference database at 99% similarity.

Phylogenetic Investigation of Communities by Reconstruction of Unobserved States 2 (PICRUSt2) was used to predict the gut microbial functional genes based on the 16S rRNA gene sequences with default settings. We then applied the linear discriminant analysis effect size (LEfSe) method with default settings to determine the gut microbial functional genes most likely to explain the differences between responders and non-responders at baseline. All statistical analyses were performed using R version 4.0.3 (R Core Team, Vienna, Austria) [11], the ggplot2 [12] and the dplyr [13] packages. *p*-values less than 0.05 were considered statistically significant.

## 3. Results

### 3.1. Characteristics of the Study Participants

There were 12 responders and 17 non-responders. All were prescribed anti-psychotic medication, and none had their antipsychotic dosage changed during the study period. In addition, none of the participants used antibiotics, took diets or supplements containing *Bifidobacterium*, or consumed a high amount of alcohol during the study period. The median age of the responders was 46 years (interquartile range, 16 years) and that of the non-responders was 41 years (interquartile range, 16 years). There were no significant differences in age between the groups (*p* = 0.49). There were 8 women (66.7%) among the 12 responders and 9 women (52.9%) among the 17 non-responders (*p* = 0.290; data not shown). The proportion of the responders and the non-responders with comorbidity of physical disease was 41.7% and 29.4%, respectively (*p* = 0.490; data not shown). Furthermore, the mean (standard deviation (SD)) of the body mass index (BMI) of the responders and the non-responders was 26.5 (6.4) and 23.6 (5.1), respectively (*p* = 0.240; data not shown). Finally, the proportion of smokers among the responders and the non-responders was 41.7% and 35.3%, respectively (*p* = 0.730; data not shown).

### 3.2. Functional Gene Compositions of the Gut Microbiota at Baseline

The gut microbial functional genes whose relative abundances were significantly different between responders and non-responders at baseline in LEfSe analysis are shown in Figure 1. Compared with non-responders, responders showed higher relative abundances of 5 functional genes included in the Kyoto Encyclopedia of Genes and Genomes (KEGG) pathway “Metabolism” (Energy metabolism, glycosyltransferases, lipid metabolism, retinol metabolism, and penicillin and cephalosporin biosynthesis), one in “Genetic Information Processing” (Protein processing in endoplasmic reticulum), and one in “Organismal Systems” (Insulin signaling pathway) (Figure 1A,B). In contrast, non-responders showed higher relative abundances of 2 functional genes included in the KEGG pathway “Metabolism” (Nucleotide metabolism and glycerophospholipid metabolism) and 2 in “Genetic Information Processing” (RNA transport and base excision repair) (Figure 1). In addition, as shown in Figure 2, we compared 14 functional genes at the same level (KEGG pathway Level 2) included in “Metabolism”. The relative abundances of the functional genes related to energy metabolism and lipid metabolism were higher in responders than in non-responders. In contrast, the relative abundances of the functional genes related to nucleotide metabolism were higher in non-responders than in responders.

## 4. Discussion

This is the first study examining the impact of the predicted functional profile of the gut microbiota at baseline on the therapeutic effects of probiotics using an interventional study in patients with mental disorder. Our results suggest that an elevated lipid and energy metabolism at baseline might be associated with the effects of probiotics on anxiety and depressive symptoms. As one potential mechanism, the end-products of lipid and energy metabolism by the gut microbiota may contribute to the maintenance of a healthy gut environment and influence anxiety and depressive symptoms associated with systemic inflammation in the host. These findings highlight the importance of assessing functional genes in the gut microbiota at baseline before probiotic therapy initiation for patients with mental disorders.

Among 11 bacterial functional genes found to have significantly different levels between responders and non-responders, “Lipid metabolism” and “Energy metabolism” are known to affect host metabolism and immune activity through their metabolites [14]. On the other hand, the other 9 bacterial functional genes play unknown roles in host metabolism and immune activity or are known to be housekeeping genes that are essential for maintaining functions in bacteria according to the KEGG. For example, glycerophospholipids are a major component of the bilayer envelope of Gram-negative bacteria and glycosyltransferases are involved in the biosynthesis of bacterial cell walls [15]. “Protein processing in endoplasmic reticulum” refers to the processing pathway in which proteins are glycosylated and folded in the endoplasmic reticulum within the bacteria, whereas “Insulin signaling pathway” is also involved in the insulin signaling pathway within bacteria. “RNA transport” is the pathway responsible for RNA transport from the bacterial nucleus to the cytoplasm, and “Base excision repair” is the major DNA damage repair pathway for processing small base lesions produced by oxidative and alkylation damage. These pathways are thus important for the maintenance of bacterial, not host, function. Therefore, of the pathways whose expression levels differed between the two groups in this study, all but Energy and Lipid metabolism are unlikely to be related to host homeostasis. Further in vitro and in vivo studies are needed to determine how these functional genes that play unknown roles in host metabolism and immune activity or that are known to be housekeeping genes influence the therapeutic response to probiotics.

The relative abundances of the functional pathways of “Lipid metabolism” and “Energy metabolism” of the gut microbiota at baseline were significantly higher in responders than in non-responders. These results might imply that the effects of *B. breve* A-1 on anxiety and depressive symptoms require sufficient lipid and energy metabolic function of the gut microbiota at baseline, although additional animal experiments and detailed mechanistic analysis are needed. The lipid and energy metabolic function of the gut microbiota has been linked to its ability to produce short-chain fatty acids (SCFAs). Gut bacteria consume and metabolize indigestible foods such as dietary fiber and mainly synthesize SCFAs as the final metabolites [16]. Gut bacteria also produce gases (CO_2_, CH_4_, H_2_) and heat, but the gross energy of SCFAs is considerably higher than that of gases and heat [17]. High production of SCFAs prevents host obesity and maintains a healthy gut environment, which could affect anxiety-depression symptoms related to systemic inflammation in the host. SCFAs are sensed by G protein-coupled receptors expressed in adipose tissue as an indicator of energy status, preventing excessive fat deposition in adipose tissue and promoting fat utilization in other tissues [18]. SCFAs have are also a major energy source for intestinal epithelial cells and to play a key role in inhibiting the growth of bad bacteria and promoting the establishment of good bacteria by lowering intestinal pH [16].

Interestingly, *Bifidobacterium* has been reported to influence the metabolism of lipids with anti-inflammatory properties, such as SCFAs and polyunsaturated fatty acids (PUFAs). Administration of *Bifidobacterium* increases the production of the SCFA butyrate by altering the relative abundance of other microbiota involved in lipid metabolism [19]. Elevated butyrate in the gut has been reported to activate regulatory T cells and thereby reduce the host’s systemic inflammation [20]. Furthermore, a higher relative abundance of *Bifidobacterium* is associated with higher levels of the PUFA docosahexaenoic acid, which is known to have anti-inflammatory properties [21]. Taken together, our results and those of these studies suggest that lipid metabolism could play an important role in the anti-inflammatory effects underlying the impact of *Bifidobacterium* on anxiety and depressive symptoms.

Evaluation and modification of the bacterial species and functional gene composition of the microbiota prior to therapy initiation may become an essential step in clinical practice to achieve maximum therapeutic efficacy. Indeed, technology for modifying the microbiota using the CRISPR-Cas system has already been established [22], and the application of this technology to clinical practice will be one of the cornerstones in the development of personalized medicine. In the field of psychiatry, where the response to treatment varies greatly from patient to patient, there are growing expectations for the evaluation of gut microbiota before therapeutic interventions and its modification.

We acknowledge that this study is subject to several important limitations. First, the functional gene analysis was performed not with shotgun metagenomic sequences, but with 16S rRNA gene sequences. One of the limitations of PICRUSt2 is that it predicts genes at the genomic level, not the transcriptional level. Therefore, what PICRUSt2 builds is not a profile of predicted functional activity, but rather a “potential” for predicted function, which needs to be interpreted with care. However, PICRUSt2, which we used to predict functional genes in the microbiota, can rigorously predict the abundance of pathways present based on a huge database of reference genomes and gene families, and the accuracy of metagenomic inference is sufficiently high [23]. Second, we did not conduct a detailed analysis of the differences in lipid and energy in particular. In the future, we would like to use metabolome analysis to measure SCFAs and lipid levels in the intestine gut and further investigate the role of SCFAs and lipid metabolism in the effects of probiotics. Third, it is unclear whether the present results can be extrapolated to depressive symptoms in patients with depression or to psychological distress in individuals without mental disorders because the study was focused on anxiety and depression in patients with schizophrenia. However, studies of gut bacteria in mental disorders have reported differences by symptom domain, regardless of differences by disease [24]. There may be a cross-disease relationship between gut bacteria and anxiety and depression, and further studies focusing on this aspect are needed.

## 5. Conclusions

In conclusion, our results indicate that elevated lipid and energy metabolism at baseline might be associated with the effects of probiotic treatment with *B. breve* A-1 on anxiety and depressive symptoms. The effect of probiotics on anxiety and depressive symptoms may require sufficient metabolic function of the gut microbiota at baseline. These findings highlight the importance of assessing functional genes in the gut microbiota at baseline before the initiation of probiotic therapy in patients with mental disorders. We believe that clinical application of the results of this study will lead to the realization of personalized medicine that maximizes the therapeutic effect on patients with mental disorders through gut microbiota analysis in the future.

## Figures and Tables

**Figure 1 jpm-11-00987-f001:**
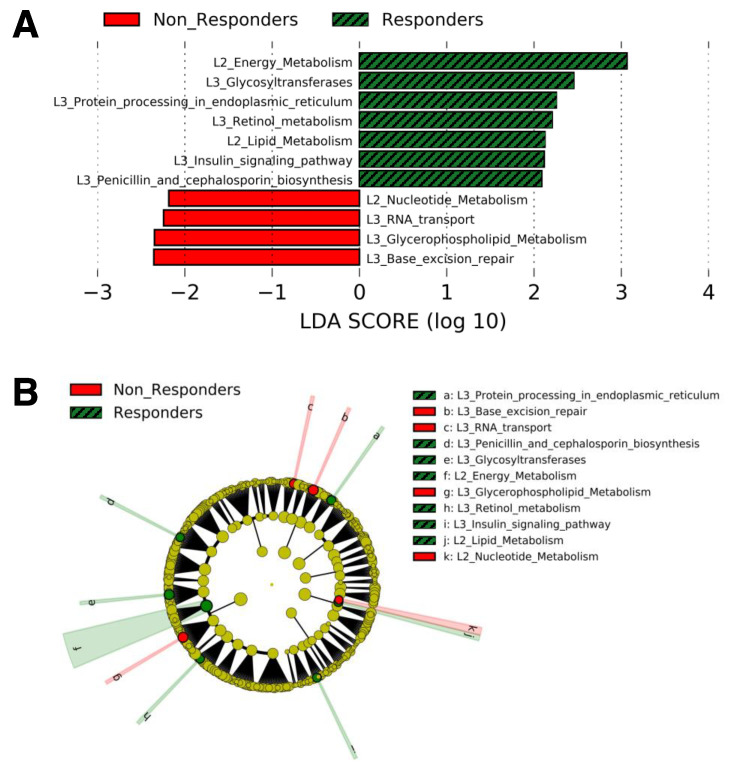
LDA scores calculated from features of the gut microbial functional genes found to exhibit different abundances between non-responders and responders at baseline. The criterion for feature selection was a log_10_ LDA score > 2.0. (**A**), Plot of pathways discovered by LEfSe ranked according to their effect size. (**B**), Cladogram representing the LEfSe results on the hierarchy. Abbreviations: L2, KEGG pathway Level 2; L3, KEGG pathway Level 3; LDA, linear discriminant analysis.

**Figure 2 jpm-11-00987-f002:**
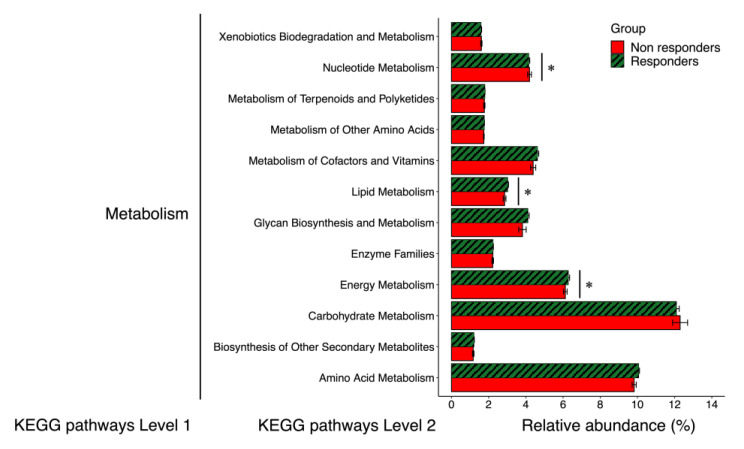
Relative abundances of functional pathways of the gut microbiota involved in metabolism. Error bars: standard error. * *p* < 0.05 on linear discriminant analysis effect size analysis (see Figure 1). KEGG, Kyoto Encyclopedia of Genes and Genomes.

## Data Availability

This study was registered in the University Hospital Medical Information Network Clinical Trials Registry (A study examining the effect of consuming foods containing probiotics on anxiety and depressive symptoms: a non-randomized and open trial, https://upload.umin.ac.jp/cgi-open-bin/ctr/ctr_view.cgi?recptno=R000029257 (accessed 29 September 2021), UMIN000025417).

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
