# Peer review of "Lipid and Energy Metabolism of the Gut Microbiota Is Associated with the Response to Probiotic Bifidobacterium breve Strain for Anxiety and Depressive Symptoms in Schizophrenia"

_jpm, 2021, doi:10.3390/jpm11100987_

Round 1
Reviewer 1 Report
Research Summary:
The authors of this manuscript use 16S functional gene predictions to investigate changes in predicted functional profiles of gut microbiota in anxiety, depressive symptoms, and schizophrenia.
General perception:
This is a generally well written manuscript covering an important topic. I do feel that it could be measurably improved by following some of the below suggestions, particularly in using more resolution of the functional pathway analysis (Level 3) and in connecting some of the actual taxa to the relevant functions.
- I am not a fan of the usage of “functional gene profile” in this manuscript. Having published quite a bit in the gene prediction space, I find I prefer labels such as “predicted functional metagenome” or “predicted functional profile” more accurate and less misleading. I would also qualify well in the discussion the benefits and limitations of 16S based functional gene predictions (the authors did address this although I feel it could be improved).
- The topic of the manuscript has tremendous merit and is novel, pertinent, and interesting.
- The idea that baseline analysis is important may seem obvious but is also often overlooked.
- I would have liked to see what bacterial taxa are responsible for the functions identified to be important.
Detailed comments and questions on sections:
Figures:
- Figures are plain but appropriate. My only comment would be that Red and Green are quite possibly the worst colors for someone who is colorblind.
- Figure 2: Differences in predicted gene abundance are not exactly striking here. Maybe Level 3 would be more informative?
Abstract:
21 I would again consider better qualification or labeling of “functional gene profile”
Introduction:
No comments.
Materials and Methods:
86 I generally like to know what packages were used in R and to see them appropriately cited.
Results:
- Was Level 3 investigated? Level 3 tends to be much more informative in my opinion.
89-91 This is a generalized descriptor of a results section. I would guess that the authors did not intend to leave it in the manuscript (?)
96-97 8 males and 9 females equate to 17 participants. 29 total participants were listed as being in groups. What about the remaining 12 participants?
Discussion:
- I would have liked some discussion on which organisms were most pertinent to the functional analysis (this information can be extracted from PICRUSt2).
- I feel the inferred connection and even speculation of the identified pathways to possible mechanisms for treatment are somewhat lacking and could be much better fleshed out.
- Another important limitation of PICRUSt2 is that you are predicting genes at the genomic level, not the transcriptional level so what you are building with PICRUSt2 is really more of a predicted function “potential” rather than a profile of predicted functional activity.
Conclusions:
- Nice and succinct conclusion. However, I do feel that it could be expanded slightly to address in more detail the importance of the finding and potential for future work.
Reviewer 2 Report
Dear Yamamura et al, it was very interesting to review your paper, this is an amazing and important field of research. In my opinion, you could improve the content of the paper by extending some sections. Please find below my comments.
- Line 42: please cite the message of the paper correctly otherwise its content is distorted, you are basically promising something which is not scientifically proven. I copy-pasted the conclusion:
”…In summary, the current evidence base for prebiotics and probiotics in the treatment of internalizing disorders appears modest. Support for the efficacy of probiotics for depression and anxiety was observed, but with generally small pooled effects. These findings are qualified, however, by the relative rarity of trials with psychiatric samples and the prevalence of non-clinical samples in the literature, which together significantly reduced the observed effects. In general, the largest effects were found for probiotics and major depression, but this should be regarded as preliminary, being limited to four trials. Future studies with clinically significant presentations are indicated and necessary adequately to evaluate the potential efficacy of prebiotics and probiotics for depression and anxiety. This is especially important given the increasing need for the development of novel psychopharmacological agents for these conditions (Hyman, 2012; Insel, 2015; Miller, 2010).”
- Materials and methods, Line 62: Maybe add when you exactly you took the fecal samples. Before administration and/or after administration of the probiotics? From the abtract I understand that you analysed the microbial community/functionality before the start of administration, is this correct?
- Line 59: consecutive outpatients means not hospitalised? How did you control confounding factors (food, drinks, medication, antibiotics, etc)?
- Results, Line 92 ff: where there any other non-/significant differences regarding functional genes between individuals besides age? Differences between male and female? Maybe include PCoAs.
- Discussion, Line 129-131: Why is this important? Would you suggest to do a fecal analysis before treatment with probiotics? Is this feasible? I would think this research aims more to the understanding of mechanism.
- Line 134: citation?
- Line 137-145: please discuss why you think this could influence the host itself.
- Line 151-153: why do you think that? That would mean B.breve A-1 only is effective in patients which actually have enough bacteria for energy metabolism (and sufficient SCFAs)? Why then does B.breve have an effect at all?
- Line 190-192: why lipids and not SCFA which are discussed as metabolites? Maybe focus then on a discussion about lipids.
Round 2
Reviewer 2 Report
Dear authors,
thanks for your revision, no further comments!